# New Developments in Imaging for Sentinel Lymph Node Biopsy in Early-Stage Oral Cavity Squamous Cell Carcinoma

**DOI:** 10.3390/cancers12103055

**Published:** 2020-10-20

**Authors:** Rutger Mahieu, Josanne S. de Maar, Eliane R. Nieuwenhuis, Roel Deckers, Chrit Moonen, Lejla Alic, Bennie ten Haken, Bart de Keizer, Remco de Bree

**Affiliations:** 1Department of Head and Neck Surgical Oncology, University Medical Center Utrecht, University of Utrecht, 3584 CX Utrecht, The Netherlands; R.Mahieu@umcutrecht.nl; 2Division of Imaging and Oncology, University Medical Center Utrecht, University of Utrecht, 3584 CX Utrecht, The Netherlands; J.S.deMaar@umcutrecht.nl (J.S.d.M.); R.Deckers-2@umcutrecht.nl (R.D.); C.Moonen@umcutrecht.nl (C.M.); 3Department of Magnetic Detection & Imaging, University of Twente, 7522 NB Enschede, The Netherlands; e.r.nieuwenhuis@utwente.nl (E.R.N.); l.alic@utwente.nl (L.A.); b.tenhaken@utwente.nl (B.t.H.); 4Department of Radiology and Nuclear Medicine, University Medical Center Utrecht, 3584 CX Utrecht, The Netherlands; B.deKeizer@umcutrecht.nl

**Keywords:** squamous cell carcinoma of head and neck, mouth neoplasms, lymphatic metastases, sentinel lymph node biopsy, diagnostic imaging, lymphatics, tracer

## Abstract

**Simple Summary:**

In early-stage (cT1-2N0) oral cancer, occult lymph node metastases are present in 20–30% of patients. Accordingly, accurate staging of the clinically negative cervical nodal basin is warranted in these patients. Sentinel lymph node biopsy has proven to reliably stage the clinically negative cervical nodal basin in early-stage oral cancer. However, due to the limited resolution of conventional sentinel lymph node imaging, occult lymph node metastasis may be missed in particular circumstances. Therefore, technical developments are necessary to bring the diagnostic accuracy of sentinel lymph node biopsy, in early-stage oral cancer, to a higher level. This review evaluates novel sentinel lymph node imaging techniques for early-stage oral cancer, such as MR lymphography, CT lymphography, PET lymphoscintigraphy and contrast-enhanced lymphosonography. Their reported diagnostic accuracy is described and their relative merits, disadvantages and potential applications are outlined.

**Abstract:**

Sentinel lymph node biopsy (SLNB) is a diagnostic staging procedure that aims to identify the first draining lymph node(s) from the primary tumor, the sentinel lymph nodes (SLN), as their histopathological status reflects the histopathological status of the rest of the nodal basin. The routine SLNB procedure consists of peritumoral injections with a technetium-99m [^99m^Tc]-labelled radiotracer followed by lymphoscintigraphy and SPECT-CT imaging. Based on these imaging results, the identified SLNs are marked for surgical extirpation and are subjected to histopathological assessment. The routine SLNB procedure has proven to reliably stage the clinically negative neck in early-stage oral squamous cell carcinoma (OSCC). However, an infamous limitation arises in situations where SLNs are located in close vicinity of the tracer injection site. In these cases, the hotspot of the injection site can hide adjacent SLNs and hamper the discrimination between tracer injection site and SLNs (shine-through phenomenon). Therefore, technical developments are needed to bring the diagnostic accuracy of SLNB for early-stage OSCC to a higher level. This review evaluates novel SLNB imaging techniques for early-stage OSCC: MR lymphography, CT lymphography, PET lymphoscintigraphy and contrast-enhanced lymphosonography. Furthermore, their reported diagnostic accuracy is described and their relative merits, disadvantages and potential applications are outlined.

## 1. Introduction

In early-stage (cT1-2N0) oral squamous cell carcinoma (OSCC), occult lymph node metastases are present in 20–30% of patients, even when the status of the regional lymph nodes has been evaluated using combinations of advanced clinical diagnostic imaging modalities (i.e., ultrasound guided fine-needle aspiration (USgFNA), magnetic resonance imaging (MRI) and/or computed tomography (CT)) [1,2,3]. As watchful-waiting in these patients has been associated with a poor prognosis, especially when compared to those in whom the clinically negative neck was electively treated [1], two strategies for the clinically negative neck in early-stage OSCC are available: elective neck dissection (END) and sentinel lymph node biopsy (SLNB) [3,4,5,6]. Although END is the strategy of choice in the majority of medical centers globally [5], which has the benefit of being a single-stage procedure, SLNB is a less invasive procedure for the 70–80% of patients without metastatic neck involvement and has overall lower morbidity rates, better quality-of-life and lower health-care costs as compared to END [7,8,9,10].

The concept of SLNB is based on the premise that lymph flow from the primary tumor travels sequentially to the sentinel lymph node (SLN) and then on to the other regional lymph nodes. Hence, the SLN is the lymph node that has the highest risk of harboring metastasis [11].

The SLNB procedure aims to identify these first draining lymph node(s), as their histopathological status reflects the histopathological status of the rest of the nodal basin. Complementary nodal treatment (e.g., surgery, radiotherapy) should be performed in case of metastatic involvement of SLN(s). A negative SLNB, however, would justify a wait-and-scan policy [12].

In short, the routine SLNB procedure consists of preoperative peritumoral injections with a technetium-99m [^99m^Tc; γ-emitter]-labelled radiotracer followed by planar dynamic and static lymphoscintigraphy including SPECT-CT (single photon emission computed tomography-computed tomography) imaging. Based on preoperative lymphoscintigraphy, the position of the SLN(s) is marked on the skin. The marked SLNs are surgically removed, using a portable γ-probe for intraoperative localization of SLNs. Subsequently, the harvested SLNs are subjected to meticulous histopathological assessment using step-serial-sectioning and immunohistochemistry [12,13,14,15].

SLNB has proven to reliably stage the clinically negative neck in early-stage OSCC with a pooled sensitivity and negative predictive value (NPV) of 87% and 94%, respectively [16]. However, an infamous limitation of the routine SLNB procedure arises in situations where SLNs are located in close vicinity of the tracer injection site. In these cases, the hotspot of the injection site can hide adjacent SLNs, which consequently hampers the discrimination between tracer injection site and SLNs (shine-through phenomenon; Figure 1). This shine-through phenomenon is particularly evident in patients with floor-of-mouth OSCC and sublingual, submental and submandibular SLNs, resulting in a significantly lower accuracy of SLNB in floor-of-mouth tumors (sensitivity 63%; NPV 90%) compared to other oral cavity subsites (sensitivity 86%; NPV 95%) [4,17,18,19,20,21].

Therefore, technical developments are needed to bring the diagnostic accuracy of SLNB for all subsites of OSCC to the same high level. This review evaluates new developments in preoperative SLN imaging techniques for early-stage OSCC: MR lymphography, CT lymphography, PET lymphoscintigraphy and contrast-enhanced lymphosonography. Furthermore, this review describes their diagnostic accuracy as reported in literature and outlines their relative merits, disadvantages and potential applications.

## 2. Results

A systematic literature search for new developments in preoperative SLN imaging techniques for early-stage OSCC resulted in a total of 452 PubMed indexed articles, of which 40 were considered relevant. Cross-reference led to 1 additional relevant study with healthy volunteers. Of these 41 articles, 27 were reviews (*n* = 1), animal or preclinical studies (*n* = 26). In particular, 20 animal or preclinical studies used similar methods for SLN identification (i.e., imaging modality, tracer) as corresponding clinical studies.

Table 1 shows the range of reported diagnostic accuracy, in terms of sensitivity and NPV, and rate of patients in which SLNs were identified using the reviewed techniques. Figure 2 illustrates how both preoperative detection and intraoperative localization of SLNs was achieved, using the reviewed techniques, as described in literature.

### 2.1. Magnetic Resonance Lymphography

Magnetic resonance (MR) lymphography with peritumoral administration of a paramagnetic gadolinium [Gd^3+^]-based contrast agent has been recently introduced in breast and cervical cancer, as an alternative method for preoperative visualization of SLNs and lymphatics [36,37,38]. These studies showed that paramagnetic gadolinium [Gd^3+^]-based contrast agents, conventionally administered intravenously for contrast-enhanced MRI or MR angiography [39], are safe and useful for peritumoral administration and SLN mapping in humans.

To review MR lymphography for SLN detection using paramagnetic gadolinium-based contrast agents in early-stage OSCC, a systematic literature search was conducted. This led to retrieval of 53 PubMed indexed articles for MR lymphography; 7 were considered relevant [22,40,41,42,43,44,45]. Of these 7 articles, 6 were animal studies [40,41,42,43,44,45]. Cross-reference led to identification of 1 relevant study with healthy volunteers [46].

In the only study that performed MR lymphography with a gadolinium-based contrast agent (i.e., gadobutrol) in OSCC patients (*n* = 26) [22], SLNs were consistently visualized in all patients and lymph node vessels were visualized in the majority of patients (81%) (Figure 3). Following MR lymphography, identified SLNs were injected with 1% patent blue dye under sonographic guidance. Subsequently, primary tumor resection and ipsilateral elective neck dissection were performed in all patients. Blue stained SLNs were dissected, marked and sent separately for histopathological assessment.

Among the 11 patients with pathologically positive necks, SLNs containing metastases were accurately identified by MR lymphography in 10 patients. In the remaining patient, MR lymphography depicted SLNs in ipsilateral neck level III. However, in the neck dissection specimen, 3 metastatic lymph nodes in ipsilateral neck level I were found, whereas no metastasis was found in level III. With histopathological assessment of the neck dissection specimen as reference standard, this approach reached a sensitivity of 90.9% with a NPV of 92.8%.

Another type of contrast agent that can be used for MR lymphography are superparamagnetic iron oxide nanoparticles (SPIO), which provide a negative contrast on MR lymphography as opposed to gadolinium-based contrast agents (Figure 4). Following peritumoral administration of SPIOs, transportation through the lymphatic system is mainly facilitated by macrophages, although unbound transport is seen as well [47]. SPIO accumulates primarily in lymph node sinuses and can be detected preoperatively on MRI and intraoperatively with a handheld magnetometer [23,47,48,49,50]. MR lymphography using SPIO has been investigated for several tumor types, including breast and prostate cancer [48,49].

The systematic literature search retrieved 116 PubMed indexed articles, of which 6 were considered relevant [23,24,25,45,50,51]. Of these 6 articles, 3 were animal studies [45,50,51]. Cross-reference did not lead to identification of additional relevant articles, resulting in a total of 3 included human studies [23,24,25]. One of these studies did not perform preoperative SPIO-enhanced MRI, but was the only study in early-stage OSCC patients that achieved intraoperative localization of SLNs with the magnetometer [23].

Mizokami et al. performed MR lymphography using SPIO in three tongue cancer patients (cT2N0), planned for tumor resection and ipsilateral elective neck dissection [24]. Seven days before surgery, patients received peritumoral injections with Resovist (Bayer Schering Pharma) of 0.1–0.3 mL, corresponding with 2.78–8.37 mg iron. MR lymphographic images were acquired at 10 min, 30 min and 24 h post-injection. On the day before surgery, [^99m^Tc]-phytate was administered peritumorally, followed by planar lymphoscintigraphy. Intraoperatively, SLNs were localized using a conventional γ-probe and were submitted for individual histopathological assessment. All SLNs depicted on 10 min MR lymphography were in accordance with planar lymphoscintigraphy and γ-probe findings. MR lymphography at 30 min and 24 h post-injection showed more uptake of SPIO in SLNs. However, MR lymphography 24 h post-injection also visualized higher echelon nodes (HEN). Besides, on MR lymphography SPIO-induced streak artifacts were seen around the injection site, but did not prevent identification of SLNs in vicinity of the tracer injection site. Histopathological assessment confirmed presence of iron in all harvested SLNs. In one patient nodal metastases were found in a harvested SLN; no additional metastases were seen in the neck dissection specimen. No follow-up results were reported in this study. In two patients, tissue swelling was observed at the injection site after administration of SPIO, which was attributed to the volume of SPIO injected.

Maza et al. evaluated fusion of lymphoscintigraphic SPECT, SPIO MR lymphography and CT, for identification of SLNs in rather complex anatomical regions [25]. Fourteen patients were included of whom two diagnosed with tongue cancer; scheduled for tumor resection and ipsilateral elective neck dissection. A mixture of [^99m^Tc]-nanocolloid and SPIO (Resovist), in total 0.5 mL, was peritumorally injected on the day before surgery. MR lymphography was acquired 2 h post-injection. Lymph nodes were assessed as SLN if they corresponded with SPECT images and exhibited signal loss on T2*-weighted sequences. SPECT-MRI fusion was successful in both OSCC patients and showed corresponding SLNs. Intraoperatively, SLNs were localized using a γ-probe and were sent for individual histopathological assessment. SLN metastases were found in the contralateral neck of one OSCC patient, leading to a complementary contralateral neck dissection. No (additional) lymph node metastases were found in the neck dissection specimens of both patients. No follow-up results were reported.

### 2.2. CT Lymphography

Another approach for high-resolution lymphography regards computed tomography (CT) lymphography using peritumoral administered iodine-based contrast agents. The use of CT lymphography has been investigated in several tumor types including breast, lung, esophageal, gastric and skin cancer [52,53,54,55,56,57,58,59,60,61,62,63]. In these studies, CT lymphography provided high-resolution visualization of SLNs, lymphatic vessels and surrounding anatomical structures.

For reviewing the application of CT lymphography in early-stage OSCC, the systematic literature search led to retrieval of 112 PubMed indexed articles for CT lymphography, of which 17 were considered relevant [26,27,28,29,30,31,40,41,64,65,66,67,68,69,70,71,72]. Of these 17 articles, 11 were animal studies [40,41,64,65,66,67,68,69,70,71,72]. Cross-reference did not lead to any additional relevant articles.

The case report of Saito et al. [31] was the first article that described the application of CT lymphography in an early-stage OSCC patient. Using CT lymphography with peritumoral injection of iopamidol (2.0 mL), a right lateral lingual lymph node was identified as SLN from a cT2N0 right oral tongue tumor. Following partial glossectomy, without any management of the neck or extirpation of the SLN, the patient showed no evidence of disease after 14 months follow-up. This case-report demonstrated that CT lymphography is suitable for visualization of small SLNs located near the primary tumor, such as lingual lymph nodes.

The first series regarding CT lymphography in early-stage OSCC patients (*n* = 31; oral tongue) was reported by Honda et al. [29]. In this study, CT images were obtained 1, 3, 5, and 10 min after administration of 1.5 mL iopamidol mixed with 0.5 mL 1% lidocaine hydrochloride. Both contrast-enhanced lymph vessels draining the tumor injection site as well as SLNs were identified in 90.3% of patients. Identified SLNs were marked for biopsy using a lattice marker, combined with intraoperative peritumoral patent blue dye injection. All patients, except for those with T1N0 OSCC and negative frozen-section assessment of SLNs (*n* = 11), underwent selective neck dissection following tumor resection. Using histopathological examination of the neck dissection specimen and a follow-up of 30 months as reference standard, this approach reached a sensitivity of 80% with a NPV of 95.8%.

In the sequel study of Honda et al., including 18 patients with cT1-2N0 oral tongue carcinoma, similar methods were used for CT lymphography, resulting in a preoperative SLN detection rate of 89% [28]. For intraoperative localization of SLNs, indocyanine green (ICG) and near-infrared imaging was used, instead of patent blue dye. In contrast to their previous study [29], only patients with advanced cT2N0 disease or positive frozen-section assessment of SLNs underwent selective neck dissection (*n* = 9). In the 16 patients with at least one detected SLN on CT lymphography, a sensitivity of 71.4% and NPV of 81.8% after 38 months median follow-up were reported.

More recently, Sugiyama et al. [30] performed CT lymphography in 20 early-stage OSCC patients. Following peritumoral administration of 2.0 mL iopamidol, SLNs and lymphatic vessels draining the injection site were detected in 95% and 90% of patients, respectively. Two lingual lymph nodes were identified as SLNs (5.4%). The optimal timing for CT scanning in this study was at both 2 and 5 min post-injection, visualizing all 37 contrast-enhanced SLNs. Intraoperative SLN detection was performed under ICG fluorescence guidance; the authors stated to have localized all CT lymphographic identified SLNs during surgery using intraoperatively administered ICG. Both number of patients with metastatic SLNs as well as follow-up results were not reported.

In the sequel study of Sugiyama et al. [26], preoperative CT lymphographic images were reevaluated in 32 early-stage OSCC patients with an approach similar to their previous study [30]. During follow-up 4/27 patients with negative SLNB (14.8%), based on CT lymphography, developed regional recurrence and 1/5 patients with SLN metastasis (20%) developed recurrence between primary tumor site and the neck. Accordingly, their approach reached a sensitivity of 55.6% and NPV of 85.2%. Reevaluation of CT lymphographic images showed a subtle increase in Hounsfield units (HU) of overlooked SLNs (*n* = 5) when compared to non-contrast CT images. Besides, their results showed that HU decreased at 10 min post-injection, indicating that iopamidol is only briefly retained in SLNs.

Figure 5 shows CT lymphographic images from a recent long-term follow-up study with early-stage OSCC patients (*n* = 27; oral tongue) [27]. In this study, SLNs were detected in 96.3% of patients using CT lymphography after peritumoral administration of 2.0 mL iopamidol. Intraoperatively, SLNs were localized using ICG and near-infrared imaging. In total, 5 patients had metastatic SLNs (18.5%) and 3 patients without SLN metastases developed regional recurrence (13.6%) after median follow-up of 76 months. This resulted in a sensitivity and NPV of 62.5% and 86.3%, respectively.

### 2.3. PET Lymphoscintigraphy

Alternatively, a potential nuclear imaging modality for improving the diagnostic accuracy of SLNB is positron emission tomography (PET). Since PET is able to detect and record a higher percentage of radioactive emitted events compared to SPECT, PET provides both higher spatial and temporal resolution (i.e., acquires higher number of frames per time unit for dynamic studies) [73]. Consequently, PET could be highly suitable for lymphoscintigraphy and may identify SLNs with higher precision than conventional lymphoscintigraphy with SPECT. Instead of a γ-emitter (e.g., [^99m^Tc], [^60^Co])-labelled radiotracer, generally used for conventional lymphoscintigraphy, PET lymphoscintigraphy requires a positron emitting isotope (e.g., [^89^Zr], [^68^Ga], [^18^F])-labelled radiotracer [74].

A systematic literature search was conducted to review PET lymphoscintigraphy in early-stage OSCC. This led to retrieval of 64 PubMed indexed articles; 4 were considered relevant [32,33,75,76]. Of these 4 studies, 1 regarded an animal study [76] and 1 a review [75] that briefly discusses results from 2 of 3 included studies in our review [33,76]. Cross-reference did not lead to any additional relevant articles.

In 2013, Heuveling et al. were the first to perform dynamic and static PET lymphoscintigraphy in 5 patients with early-stage OSCC, following peritumoral administration of zirconium-89 [^89^Zr]-labelled nanocolloid [33]. Subsequently, 7–9 days after PET lymphoscintigraphy, the routine SLNB procedure with [^99m^Tc]-labelled nanocolloid was performed. The results of PET and SPECT lymphoscintigraphy were compared. PET lymphoscintigraphy was able to visualize all foci (*n* = 22) that were identified on SPECT-CT and even visualized 5 additional foci that were considered to be SLNs; all of which were located near the injection site of the primary tumor (Figure 6). Of these 5 additional foci, considered to be SLNs, 2 regarded lingual lymph nodes. Furthermore, in 4 patients (80%) lymphatic vessels were visualized on dynamic PET lymphoscintigraphy. Intraoperatively, the additionally visualized PET foci close to the injection site could not be localized with the conventional portable γ-probe, due to difficulties in differentiating between SLN and injection site. In two patients metastatic SLNs were found, follow-up results were not reported.

In their sequel study, Heuveling et al. achieved both preoperative SLN detection with PET lymphoscintigraphy, as well as intraoperative SLN localization using a handheld PET-probe, after peritumoral administration of [^89^Zr]-labelled nanocolloid [32]. This study included 5 OSCC patients who underwent tumor resection including neck dissection (i.e., clinically N1 disease or access of the neck was required for tumor resection or flap reconstruction). Preoperatively 13 SLNs were identified by PET lymphoscintigraphy, whereas the PET-probe detected 10 of 13 SLNs intraoperatively (77%). In this population, 3 patients (60%) had nodal metastases; in 1 patient the histopathologically positive SLN, found in the neck dissection specimen during histopathological assessment, was not localized with the PET-probe, although it was depicted on preoperative PET lymphoscintigraphy. None of the patients developed locoregional recurrence after a median follow-up of 25 months. With histopathological examination of the neck dissection specimen and follow-up as reference standard, this approach reached a sensitivity of 67% with a NPV of 67%. The authors concluded that PET lymphoscintigraphy using [^89^Zr]-labelled nanocolloid may improve preoperative SLN detection, although it should be combined with other tracers for intraoperative localization.

### 2.4. Contrast-Enhanced Lymphosonography

In contrast-enhanced ultrasound (CEUS), echogenic particles such as microbubbles are administered to obtain information on vascularization or delineation of body cavities during ultrasound (US) imaging. FDA and EMA approved microbubbles consist of perfluorocarbons or sulfur hexafluoride (SF_6_) gas surrounded by a thin biocompatible shell generally made of phospholipids or proteins [77,78]. Due to their compressibility and the large difference in acoustic impedance between gas and the surrounding liquid (i.e., blood or lymph) they strongly scatter ultrasound pulses. In addition, due to nonlinear microbubble oscillations, the scattered signal contains higher harmonic frequencies. These higher harmonic frequencies can be distinguished from the fundamental frequency scatter emitted by relatively incompressible tissue surroundings, consequently enhancing microbubble containing structures [77,78,79] (Figure 7).

Microbubbles are typically administered intravenously, but have more recently been proposed as a radiation-free tracer for lymphosonography. In breast cancer, studies reported SLN localization rates between 60–100%. For CEUS-guided SLNB a pooled sensitivity of 54% (95% CI 47–61%) and a NPV of 83–92%, were reported [80]. Few to no adverse events of the procedure were registered; any minor adverse events consisted of localized redness, pain or bruising at the injection site [80,81].

To review contrast-enhanced lymphosonography in OSCC, a systematic literature search was conducted, which led to retrieval of 107 PubMed indexed articles. A total of 6 studies were considered relevant (i.e., 2 clinical studies [34,35] and 4 large animal studies [82,83,84,85]). Cross-reference did not lead to identification of additional relevant articles.

Figure 7 illustrates the procedure used in the two clinical studies [34,35]. Gvetadze et al. [34] used sulfur hexafluoride (SF_6_) phospholipid microbubbles (SonoVue; Bracco International B.V.) in 12 patients with T1-2cN0 oral tongue carcinoma and looked for lymph node enhancement after repetitive peritumoral injections. Fifteen SLNs were identified in 11/12 patients (91.7%). No attempt was made at intraoperative localization of identified SLNs and therefore the correlation between identified SLNs and histopathological assessment was lacking. In the second clinical study, Wakisaka et al. [35] studied lymphosonography with perfluorobutane phospholipid (C_x_F_y_) microbubbles (Sonazoid; GE Healthcare, UK) in 10 patients with T1-4N0 oral or oropharyngeal carcinomas. Sonazoid was injected in four peritumoral locations. In 8/10 patients, 12 SLNs were identified. In one patient with a T4 tumor, Sonazoid had to be injected intratumorally and no SLNs were identified. SLN locations were marked on the skin. The next day indigo carmine blue dye was injected intraoperatively at the same injection sites. All lymph nodes marked during lymphosonography, which were not always dyed blue, were examined with frozen section analysis. Since frozen section analysis was negative in all cases, a less extensive neck dissection was performed. No metastatic lymph nodes were found during histopathological examination of neck dissection specimens. No follow-up results were reported for both studies. Contrast-related adverse events did not occur [34,35].

## 3. Discussion

This paper reviewed new developments in preoperative SLN imaging techniques in patients with early-stage OSCC. None of the included clinical studies contradicted outcomes or clinical translation predictions from corresponding animal studies, in regard of SLN identification using these novel techniques [22,23,24,25,26,27,28,29,30,31,42,43,44,45,46,50,51,64,66,67,69,70,71,72]. The overall reported rate of patients in which SLNs were identified using the presented techniques ranged from 89–100%. The overall reported sensitivity ranged from 56–91%, with a NPV of 67–96% (Table 1).

Although the diagnostic accuracy of most presented techniques appears to be inferior to conventional lymphoscintigraphy including SPECT-CT, there are several promising advantages to the presented preoperative SLN imaging techniques which will be discussed individually in the subsections below. Accordingly, drawbacks of the presented techniques and methodology of the included studies will be discussed as well. While other (head and neck) tumor sites were not included in this review, the discussed strengths and flaws of performing SLNB using these techniques may also be relevant to other (head and neck) tumor sites. A summary of relative merits and disadvantages for each technique is listed in Table 2.

### 3.1. MR Lymphography

Bae et al. showed that MR lymphography using gadobutrol, is a promising technique for SLN detection in early-stage OSCC, with a sensitivity of 90.9%, a NPV of 92.8% and lymphatic vessel visualization in 81% of patients [22].

The high spatial resolution, high signal-to-noise ratio and few artifacts that MR lymphography with gadolinium-based contrast agents provides, even when compared to MR lymphography with SPIO, is the foremost asset of this technique [24,48,86,87,88]. These features result in accurate anatomical detail and facilitate visualization of lymphatic vessels, which is helpful in assessing whether a contrast-enhanced lymph node is a true SLN or a higher echelon node (HEN) [33]. Moreover, the high spatial resolution of MR lymphography eliminates the shine-through phenomenon, allowing identification of SLNs in vicinity of the tracer injection site. Additionally, MR lymphography is free of radiation exposure and does not require radioisotopes, which is of particular benefit if specific nuclear medicine facilities are unavailable [89,90].

Nevertheless, the low molecular weight of gadolinium-based contrast agents result in rapid lymphatic transportation, little retention in SLNs and rapid washout of the contrast agent [46,91]. This could increase the risk to overlook SLNs and of contrast-enhanced HEN(s) to erroneously being considered SLN(s). Since Bae et al. performed elective neck dissection in all patients, used only histopathological examination of the neck dissection specimen as reference standard and did not report any follow-up results (e.g., nodal recurrence), it is uncertain whether SLNs were overlooked with this technique [22]. Therefore, the diagnostic accuracy of SLNB using MR lymphography with gadolinium-based contrast agents in early-stage OSCC patients is yet to be established in larger studies with histopathological examination and follow-up as reference standard.

In addition, MR lymphography with gadolinium-based contrast agents cannot be performed when MRI or administration of these agents is contraindicated [89]. Besides, gadolinium-based contrast agents are not registered for lymphography and clinical trials on MR lymphography using these contrast agents are required before this technique can be implemented in routine clinical care.

Moreover, it is important to note that gadolinium-based contrast agents cannot be detected intraoperatively. The solution offered by Bae et al. [22], i.e., injection of identified SLNs with blue dye, is probably not reliable enough to assess whether the observed SLNs depicted on MR lymphography exactly matched the same nodes in the neck dissection specimen. A proposed alternative for intraoperative localization of SLNs is fluorescence guided surgery following peritumoral injection of ICG [92,93]. However, due to limited tissue penetration of the fluorescent signal and rapid flow through lymphatics of unbound ICG, matching of preoperative depicted SLNs and intraoperative fluorescent lymph nodes is challenging [90].

The inability to detect gadolinium intraoperatively, may be overcome by using SPIO for MR lymphography, as SPIO can be detected by both MRI and a handheld magnetometer [23,24]. Accordingly, SPIO may facilitate intraoperative localization of preoperative depicted SLNs, while maintaining benefits of MR lymphography over other imaging modalities (Table 2). Still, a correlation between preoperatively identified SLNs on MR lymphography and intraoperative localized SLNs with the handheld magnetometer has not yet been reported for early-stage OSCC patients.

The first results of MR lymphography using SPIO are auspicious, as all identified SLNs by MR lymphography corresponded with those identified by conventional lymphoscintigraphy [24,25]. Besides, adequate differentiation of SLN from injection site was seen [24] and precise anatomical information on SLN location was acquired when fused with SPECT [25].

However, some challenges for MR lymphography with SPIO remain. First of all, both ideal SPIO particle size and amount of iron administered are still under consideration. A hydrodynamic diameter of 59 nm was considered most suitable due to its fast uptake in lymphatics, retention in SLNs and its high accumulation when compared to SPIOs with a hydrodynamic diameter of 32 nm and 111 nm [50]. Hence, Resovist (45–60 nm) and Magtrace (59 nm) may be fitting candidates [23,24,25]. With respect to the volume of SPIO administered with corresponding iron quantity, a considerable difference is seen among reports [23,24,25]. While a higher concentration may assist intraoperative localization of SLNs [23], excessive concentrations of SPIO can lead to disproportionate signal voids on MR lymphography and may hamper preoperative SLN identification [87]. Vice versa, a lower concentration may benefit preoperative SLN identification [24], but may impede intraoperative localization [48]. Furthermore, the negative contrast that SPIO provides on MR lymphography, which can be induced by other factors as well (i.e., dental implants, tissue interfaces, background noise, air), may confound the efficiency of detecting SLNs [25,87,88]. Moreover, in regard of intraoperative localization of SPIO-enhanced SLNs using the magnetometer, magnetic signals deriving from metal elements (e.g., pacemakers, prosthetics, surgical instruments) interfere with the magnetometer. This can instigate some logistical issues, such as requiring plastic surgical instruments, and can even lead to a contraindication for using the magnetometer in some cases (e.g., patients with pacemakers or prosthetics) [23]. Finally, concerns were addressed concerning swelling, local inflammation and pain of the injection site following administration of SPIO, which may depend on the volume of SPIO administered [23,24].

Some reports mention a higher number of identified SLNs on MR lymphography with SPIO when compared to conventional lymphoscintigraphy and SPECT-CT, due to the better resolution of MR lymphography [48,49]. It can be debated if the higher number of identified SLNs by MR lymphography with SPIO includes not only true SLNs, but HENs as well. Since Mizokami et al. showed more enhanced lymph nodes at 24 h post-injection, which were considered HENs, the timing of MR lymphography following SPIO administration seems to be pertinent in selecting the right contrast-enhanced lymph nodes for SLNB [24]. To distinguish true SLNs from HENs, visualization of lymphatic vessels may provide a solution. However, visualization of contrast-enhanced lymphatic vessels was not reported in any of the included studies [24,25]. To enable visualization of contrast-enhanced lymphatic vessels administration of smaller SPIOs is suggested, but is criticized by their faster migration through the lymphatic system [24].

Currently, the limited number of early-stage OSCC patients who underwent MR lymphography with SPIO prevents assessment of its diagnostic accuracy. Larger studies with adequate reference standards (i.e., histopathological assessment including follow-up) should be conducted to establish the diagnostic accuracy of MR lymphography with SPIO in OSCC patients.

In conclusion, MR lymphography using gadolinium-based contrast agents may currently not offer an alternative for conventional SLNB using radiotracers, mainly due to the lack of reliable intraoperative localization of preoperatively depicted SLNs. MR lymphography with SPIO may provide a solution, as it allows for intraoperative localization of SLNs with a magnetometer. However, MR lymphography with SPIO is subject to other limitations that may confound the efficiency of preoperative detection and intraoperative localization of SLNs. Nonetheless, MR lymphography using either contrast agent can provide precise preoperative anatomical localization and identification of SLNs, particularly in situations with close spatial relation between injection site and SLN(s). Accordingly, MR lymphography might be a valuable addition to radiotherapy planning (e.g., higher radiation dose on SLNs), by performing MR lymphography as part of MRI that is increasingly used for radiotherapy planning in head and neck cancer [94]. MR lymphography-guided nodal irradiation may improve regional control, reduce acute and late radiation-related toxicity and enhance health-related quality-of-life [95].

### 3.2. CT Lymphography

CT lymphography has been proposed as a high potential alternative for conventional lymphoscintigraphy, with a sensitivity ranging from 56–80% and a NPV ranging from 82–96% [26,27,28,29]. Two series reported enhanced lymphatic vessel visualization in 90% of their patients [29,30]; in two studies lingual lymph nodes were identified as SLNs using CT lymphography [30,31].

CT lymphography shares several beneficial properties with MR lymphography: high spatial resolution, visualization of lymphatic vessels and elimination of shine-through phenomenon. The latter has been demonstrated by the identification of lingual lymph nodes as SLNs using CT lymphography [30,31]. Besides, CT lymphography does not require specific nuclear facilities and is easily implemented due to the wide availability of CT and iodine-based contrast agents [28,89]. Compared to MRI, CT has lower costs and is considered more comfortable for patients [89].

Yet, challenges for CT lymphography are similar to those in MR lymphography using gadolinium-based contrast agents. First of all, iodine-based contrast agents cannot be detected intraoperatively. Most authors used fluorescence guidance with intraoperatively administered ICG for SLN localization of preoperatively depicted SLNs by CT lymphography [26,27,28,30]. As previously mentioned, matching of preoperative depicted SLNs and intraoperative fluorescent lymph nodes is challenging [90]. Secondly, the rapid lymphatic transportation, limited retention in SLNs and rapid washout of iopamidol increases the risk to overlook SLNs and of contrast-enhanced HEN(s) to erroneously being considered SLN(s). This risk has been especially emphasized by Sugiyama et al., who showed that overlooked SLNs were only marginally contrast-enhanced on CT lymphography and that iopamidol was only briefly retained in SLNs [26].

Additional challenges arise for CT lymphography, especially when compared to MR lymphography, since CT has poor soft tissue contrast and is prone to artefacts from dental amalgam or orthopedic material, if present, which may hamper adequate visualization of SLNs. Besides, CT implies radiation exposure and, although only a low volume (2 mL) is used compared to regular intravenous use, iodine-based contrast agents may induce anaphylactic reactions, contrast-induced nephropathy or thyroid dysfunction [89]. However, contrast-related adverse events did not occur in any of the included studies [26,27,28,29,30,31].

Further developments regarding CT lymphography should address these limitations (i.e., dual-tracer methods, high velocity lymphatic drainage tracer, limited retention of tracer in SLNs) to improve its diagnostic accuracy for SLNB.

As an alternative for iopamidol as CT lymphographic tracer, lipiodol might be worth considering. In contrast to iopamidol, lipiodol is oil-based with higher viscosity and is registered and widely used for lymphographic purposes [96]. The higher viscosity of lipiodol might result in increased retention in SLNs and delayed tracer wash-out, possibly improving preoperative SLN detection on CT lymphography. Moreover, lipiodol has been combined with ICG as a single emulsion, which could overcome the limitations of dual tracer methods, potentially enabling reliable intraoperative localization of preoperative depicted SLNs [97]. This has yet to be investigated in a clinical trial with an adequate reference standard (i.e., histopathological examination and follow-up).

Although CT lymphography requires some further developments, it has the potential for highly accurate identification of SLNs in early-stage OSCC patients. Especially in cases where SLNs are in close vicinity to the tracer injection site. Besides, analogous to MR lymphography, CT lymphography, performed concomitantly with conventional CT imaging for radiotherapy planning, may facilitate more targeted radiotherapy and consequently improve regional control, reduce acute and late radiation-related toxicity and enhance health-related quality-of-life [94,95].

### 3.3. PET Lymphoscintigraphy

Heuveling et al. demonstrated the high potential of preoperative PET lymphoscintigraphy using [^89^Zr]-labelled nanocolloid for SLN detection in OSCC patients, by visualizing all foci identified on SPECT-CT and even detecting 5 additional SLNs in vicinity of the tracer injection site. Additionally, in 80% of patients, lymphatic vessels were visualized and 2 lingual lymph nodes (7%) were identified as SLNs [33].

In correspondence with MR-and CT lymphography, the high spatial resolution of PET lymphoscintigraphy enables identification of SLNs located close to the tracer injection site, which was demonstrated by detection of 2 lingual SLNs using PET lymphoscintigraphy. Moreover, PET lymphoscintigraphy provides both high temporal resolution as well as visualization of lymphatic vessels, contributing to better differentiation between true SLNs and HENs [33].

In contrast to the other presented techniques in this review, PET lymphoscintigraphy permits the use of commonly administered tracers for SLNB (e.g., nanocolloids, tilmanocept), whose kinetics have proven to be particularly suitable for SLNB [14,98]. Moreover, Heuveling et al. achieved reliable intraoperative localization of SLNs that were preoperatively identified by PET lymphoscintigraphy, using a handheld PET-probe [32]. Consequently, this method is unaffected by limitations of dual tracer methods.

Although intraoperative localization of SLNs using a handheld PET-probe was considered feasible, some concerns were addressed [32]. First of all, the PET-probe detected only 12/15 SLNs as identified by PET lymphoscintigraphy, which was attributed to the PET-probe’s limited sensitivity, resulting in a relatively low accuracy of the procedure (i.e., sensitivity 67%; NPV 67%). Secondly, a handheld PET-probe is relatively large in size because of features necessary for detection of high-energy photons from positron emitting isotopes [32,99]. Due to the limited sensitivity and large size of the PET-probe, wider skin incisions and exploration of the neck were required for SLN localization [32].

To overcome the problems with the use of a PET-probe, a radiotracer labelled with both [^89^Zr] and a γ-emitter (e.g., [^99m^Tc]) could allow high-resolution preoperative PET lymphoscintigraphy and intraoperative localization of SLNs using the conventional portable γ-probe. However, due to its half-life of 78.4 h, [^89^Zr] will interfere with the [^99m^Tc]-signal [100]. Therefore, a positron emitting isotope with a shorter half-life is required to enable detection of the [^99m^Tc]-signal for intraoperative localization of SLNs using the conventional portable γ-probe.

Fluorine-18 [^18^F] is considered the ideal radioisotope for PET imaging owing to the low positron energy (0.64 MeV), providing high-resolution images. Furthermore, [^18^F] has a half-life of only 110 min [101]. However, [^18^F] relies on C−F bond formation and is therefore difficult to label to currently used radiotracers for SLNB (e.g., nanocolloids or tilmanocept) [102]. Recently, PET lymphoscintigraphy with interstitially injected [^18^F]-FDG has been investigated in patients with cervical or endometrial cancer and in healthy subjects [103,104]. Hypothesized was that the small size of the tracer allows passage through channels infiltrated with tumor cells, and that its molecular function allows uptake by tumor cells, which is not achieved by any of the currently used radiotracers for SLNB [103]. In the study with cervical or uterine cancer patients, SLN mapping was successful in 80% of patients [103]. In the study with healthy subjects however, PET lymphoscintigraphy using [^18^F]-FDG was not considered feasible due to significant tracer washout to systemic capillaries [104].

Alternatively, gallium-68 [^68^Ga] is a good candidate due to its half-life of only 68 min and its production with a [^68^Ga]-generator, which provides an opportunity to prepare PET-radiopharmaceuticals on site when needed [100,105]. Moreover, its chemical properties allow labelling to various diagnostic molecules [106].

Whereas labelling of nanocolloids with [^68^Ga] is complicated, mainly due to instability of the bond between [^68^Ga] and nanocolloids [106], [^68^Ga] has been successfully labelled to tilmanocept [107]. Moreover, fluorescent (IRD-800CW)-labelled tilmanocept can be radiolabelled with both [^68^Ga] and [^99m^Tc]. The resulting tri-modal agent provides high-resolution preoperative PET-images for SLN mapping and intraoperative localization of SLNs with both a conventional portable γ-probe and fluorescence imaging [108]. This tri-model agent has been successfully tested with reliable SLN identification in animal models [109,110]. Although PET lymphoscintigraphy using this tri-model agent might provide a solution to the issues addressed for SLNB in early-stage OSCC, it is indisputable that first it has to be investigated in a clinical trial with adequate reference standards.

### 3.4. Contrast-Enhanced Lymphosonography

Compared to conventional lymphoscintigraphy, lymphosonography has many advantages (Table 2). Importantly, microbubbles are free of ionizing radiation and have a good safety profile, which was extensively documented for intravenous administration [111,112,113]. Secondly, lymphosonography is not affected by the shine-through phenomenon. Furthermore, none of the studies in humans or large animals found HEN enhancement [34,35,82,83,84,85]. It is possible that this is prevented by phagocytosis of microbubbles (which was histologically confirmed in animals for Sonazoid [85]) and the size of microbubbles compared to small-molecule dyes. Another advantage is the possibility to use lymphosonography preoperatively to improve lymph node selection for USgFNA. The sensitivity of USgFNA alone ranges from 45 up to 90% [114,115]. Adding lymphosonography could lead to more true positive patients, in whom the complex SLNB procedure may be omitted. A clinical trial using the combination of lymphosonography and USgFNA preceding SLNB will have to determine the value of this technique in head and neck oncological practice. Finally, ultrasound equipment is globally widely available and its mobility provides the option to use it in the operating room. Accordingly, lymphosonography may extend the application of SLNB from OSCC to less reachable sites of the head and neck (i.e., nasopharynx, oropharynx, larynx and hypopharynx), by allowing both peritumoral injection as well as SLN identification under general anesthesia.

However, lymphosonography has some disadvantages (Table 2). Foremost, the procedure is highly operator dependent and fast (a few seconds to minutes) transportation of microbubbles through the lymphatic system can make SLN identification challenging. Therefore, experienced staff will have to be appointed and trained. Future research will need to determine interobserver variability. Furthermore, if used without FNA it might be challenging to intraoperatively localize SLNs identified with preoperative lymphosonography; a reliable system to mark the exact location of SLNs is necessary. This drawback however is valid for several preoperative SLNB imaging techniques (i.e., CT lymphography, MR lymphography), and can be circumvented by combining lymphosonography with USgFNA or by performing lymphosonography intraoperatively.

Besides, further research is needed to find out which CEUS imaging method and which microbubble are most suitable. The two clinical studies report a specific contrast imaging mode at a low mechanical index (MI), thus leaving the microbubbles intact [34,35,82,83,85]. Four animal studies performed lymphosonography in the head and neck region using Sonazoid, combined with either blue dye or ICG, in swine and rabbits without tumors [82,83,85] and with Definity in dogs with spontaneously arisen tumors [84]. The studies in swine added color flow Doppler at a high (microbubble destructing) mechanical index of 1.0 to confirm the presence of microbubbles [82,83]. In dogs power Doppler with a mechanical index of 1.3 was used primarily, which produces a color flair upon microbubble destruction [84]. To select the most suitable microbubble, it is necessary to consider practicalities: using a microbubble that quickly reaches SLNs and is retained and detectable in the SLN for a long time might increase reproducibility. SonoVue consists of SF_6_ phospholipid microbubbles with a mean bubble diameter of ~2.5 μm [79], while Sonazoid consists of perfluorobutane phospholipid (C_x_F_y_) microbubbles with a mean bubble diameter ~2.1 μm [116]. In most studies the time between peritumoral administration and lymph node enhancement (transit time) was described. Although no within-study comparisons have been made and clinical studies cannot be compared directly to preclinical studies, the transit time appears to be shorter for SonoVue (10–50 s post-injection [34]), than for Sonazoid (1–11 min post-injection [82,83,85]). Sonazoid enhancement seems to persist longer, namely ≥90 min [85], versus 2–4 min with SonoVue [34]. This could explain why multiple injections were necessary to identify SLNs in the clinical study using SonoVue [34]. However, Sonazoid has not yet been approved by FDA and EMA as a US contrast agent, which could complicate its application in clinical lymphosonography trials.

To conclude, lymphosonography is a promising method, but current clinical experience in OSCC is sparse. The two published clinical studies indicate that this technique is feasible, with SLN detection rates of 80 and 92% [34,35]. Unfortunately, correlation with histopathology is still lacking: in the only study that attempted this, no metastatic lymph nodes were detected [35]. Larger studies, preferably with histopathological examination and follow-up as reference standard, are needed to determine the diagnostic accuracy (i.e., sensitivity and NPV) of this technique for SLNB in OSCC and its place in the diagnostic workflow.

## 4. Materials and Methods

A systematic literature search for relevant English written literature published up to 25 May 2020 was conducted in the PubMed database. Search syntaxes combined synonyms and medical subject headings (MeSH) terms for both OSCC and SLNB and was performed for all imaging techniques separately (i.e., MR lymphography, CT lymphography, PET lymphoscintigraphy and contrast-enhanced lymphosonography). Subsequently, title and abstract screening was performed by four authors (R.M, J.S.d.M., E.R.N and R.d.B.). The reference lists of included studies were screened to identify any additional relevant publications. No critical appraisal of the selected literature was performed. This review adheres to the PRISMA guidelines [117].

### 4.1. MR Lymphography

The following keywords and MeSH terms were included for MR lymphography: (“Mouth”[MeSH]) or (“Oral”) or (“Head and Neck”) and (“Sentinel lymph node”[MeSH]) or (“Lymph”) and (“Node”) or (“Sentinel”) and (“Node”) or (“Sentinel node”) and (“Lymphography”[MeSH]) or (“Lymphography”) or (“Lymphangiography”) and (“Magnetic resonance imaging”[MeSH] or (“Magnetic”) and (“Resonance”) and (“Imaging”) or (“Magnetic resonance imaging”) or (“MRI”) or (“MR”).

For magnetic detection of SLNs using superparamagnetic iron oxide, the following keywords and MeSH terms were included: (“Mouth”[MeSH]) or (“Oral”) or (“Head and Neck”) and (“Sentinel lymph node”[MeSH]) or (“Lymph”) and (“Node”) or (“Sentinel”) and (“Node”) or (“Sentinel node”) and (“Iron”[MeSH]) or (“Iron oxide”) or (“SPIO”) or (“SPION”) and (“Magnetics”[MeSH] or (“Magnetic”) or (“Superparamagnetic”) or (“superparamagnetic iron oxide”).

### 4.2. CT Lymphography

The following keywords and MeSH terms were included for CT lymphography: (“Mouth”[MeSH]) or (“Oral”) or (“Head and Neck”) and (“Sentinel lymph node”[MeSH]) or (“Lymph”) and (“Node”) or (“Sentinel”) and (“Node”) or (“Sentinel node”) and (“Lymphography”[MeSH]) or (“Lymphography”) and (“CT”) or (“Computed Tomography”) or (“Computed”) or (“Tomographic”).

### 4.3. PET Lymphoscintigraphy

The following keywords and MeSH terms were included for PET lymphoscintigraphy: (“Mouth”[MeSH]) or (“Oral”) or (“Head and Neck”) and (“Sentinel lymph node”[MeSH]) or (“Sentinel lymph node”) or (“Sentinel”) and (“Node”) or (“Sentinel node”) and (“Positron Emission Tomography Computed Tomography”[MeSH]) or (“Positron-Emission Tomography”[MeSH]) or (“PET”) or (“Positron”) or (“PET/CT”) or (“PET-CT”).

### 4.4. Contrast-Enhanced Lymphosonography

The following keywords and MeSH terms were included for contrast-enhanced ultrasound lymphography: (“Mouth”[MeSH]) or (“Oral”) or (“Head and Neck”) and (“Sentinel lymph node”[MeSH]) or (“Sentinel lymph node”) or (“Sentinel”) and (“Node”) or (“Sentinel node”) and (“Contrast-enhanced”) or (“Contrast-assisted”) or (“CEUS”) or (“Microbubbles”) or (“Sonovue”) or (“Sonazoid”) or (“Optison”) or (“Levovist”) or (“Imagent”) or (“Imavist”) or (“Definity”) and (“Diagnostic Imaging”) or (“Diagnostic”) and (“Imaging”) or (“Ultrasound”) or (“Ultrasonography”[MeSH]) or (“Ultrasonography”) or (“Ultrasonics”[MeSH]) or (“Ultrasonics”).

## 5. Conclusions

Novel diagnostic imaging techniques for detection of SLNs have the potential to bring the diagnostic accuracy of SLNB to a higher level for all early-stage OSCC subsites. However, technical improvements and further research of these novel techniques are required, if they are to replace the conventional SLNB procedure with [^99m^Tc]-labelled radiotracers. Nevertheless, several of these novel techniques may already become valuable by facilitating more targeted radiotherapy; adjusting the radiation dose based on the tumor’s individual lymphatic drainage pattern.

## Figures and Tables

**Figure 1 cancers-12-03055-f001:**
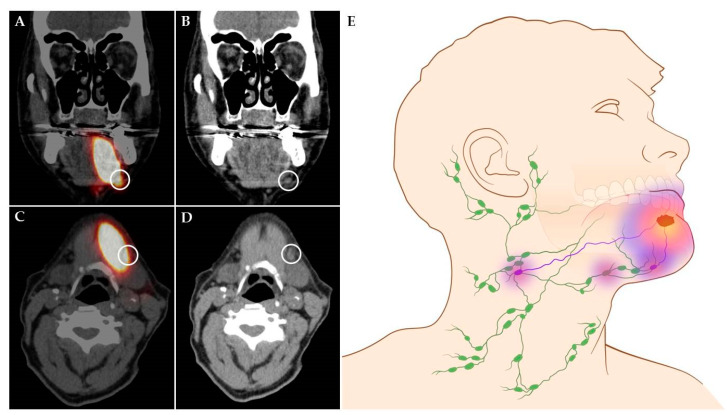
Shine-through phenomenon in 72-year-old patient with a cT1N0 floor-of-mouth carcinoma. (**A**,**C**) Coronal and axial SPECT-CT images: radiation flare of the tracer injection site over shines a sentinel lymph node located in cervical lymph node level Ib (white circle). (**B**,**D**) Coronal and axial low-dose CT images of same patient: (sentinel) lymph node located in cervical lymph node level Ib that could not be differentiated from the hotspot originating from tracer injection site on SPECT-CT (white circle). (**E**) Schematic illustration of shine-through phenomenon. (**A**–**D**) Informed consent has been obtained from this patient. (**E**) ©University Medical Center Groningen.

**Figure 2 cancers-12-03055-f002:**
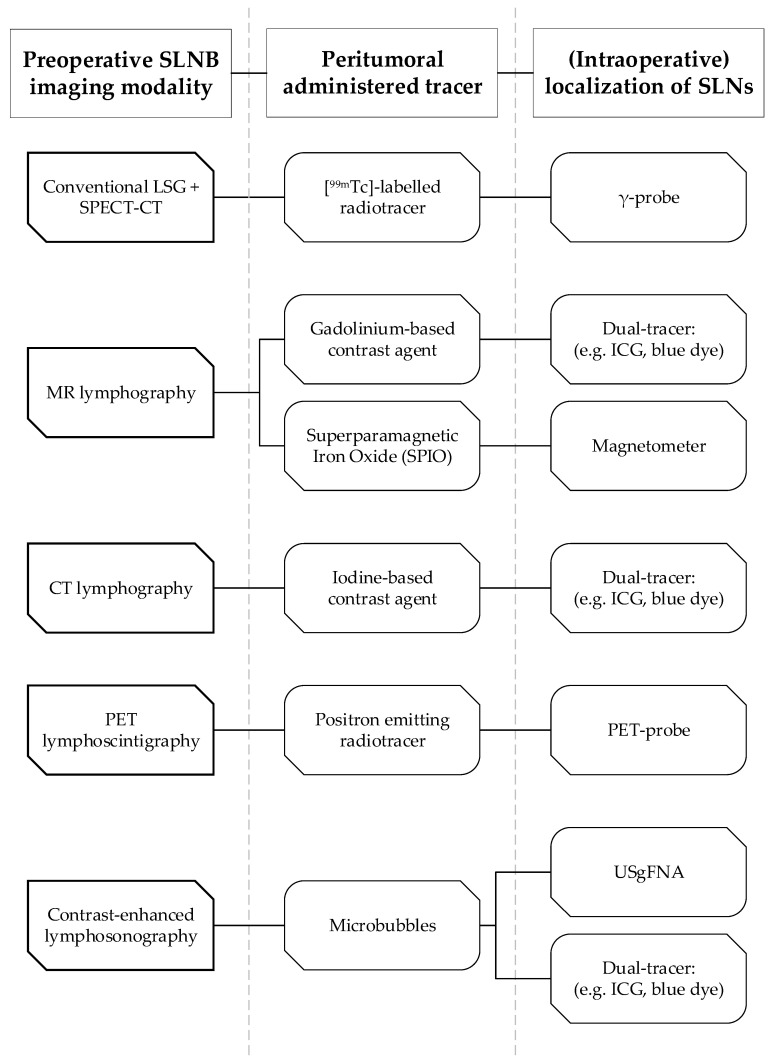
Overview of the reviewed preoperative SLN imaging techniques (column 1), the administered tracers for the corresponding techniques (column 2) and their intraoperative SLN localization techniques (column 3) as described in literature. SLNB; sentinel lymph node biopsy, LSG; lymphoscintigraphy, SPECT-CT; single photon emission computed tomography-computed tomography, MR; magnetic resonance, CT; computed tomography, PET; positron emission tomography, ICG; indocyanine green, USgFNA; ultrasound guided fine needle aspiration.

**Figure 3 cancers-12-03055-f003:**
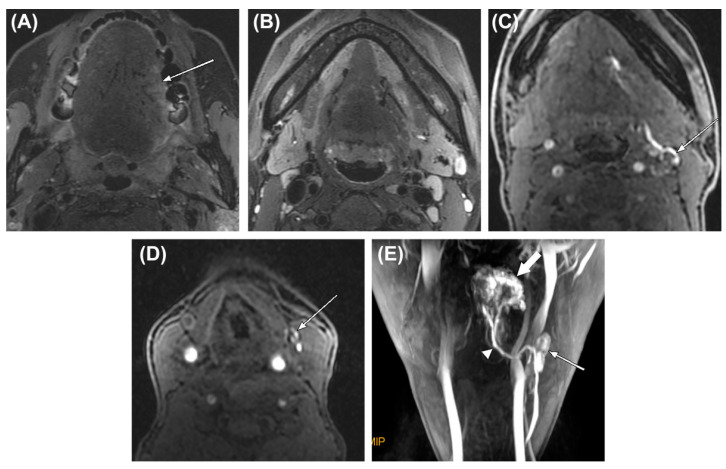
A 38-year-old woman with oral tongue cancer and palpably negative neck. (**A**,**B**) Fat-saturated T2-weighted MRI scans show a shallow infiltrative tumor on the left lateral surface of oral tongue (arrow) and several small lymph nodes in the submandibular areas. (**C**,**D**) After peritumoral injection of contrast, MR lymphography revealed two first-enhanced lymph nodes in left level IB and IIA (arrows) on the first phase of the dynamic scan, respectively. (**E**) The maximum intensity projection reconstruction image of MR lymphography shows the contrast injection site in the tongue (thick arrow), the assumed sentinel lymph node (thin arrow), and the lymph vessel connecting them (arrowhead). After neck dissection, the assumed sentinel lymph nodes observed on MR lymphography revealed no metastasis on histologic examination [22]. Figure used with permission of John Wiley and Sons©, permission license number 4807630108259.

**Figure 4 cancers-12-03055-f004:**
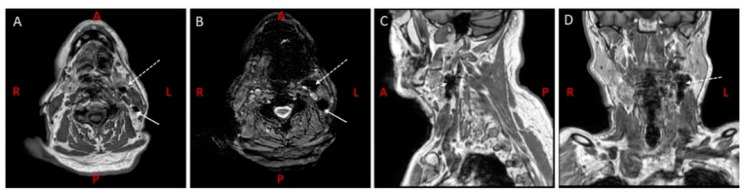
MR lymphography using superparamagnetic iron oxide nanoparticles in a 77-year-old man with oral tongue cancer and a clinically negative neck. (**A**) T1-weigted 3D fast-field echo (FFE) show uptake of SPIO in two SLNs in level IIa (dotted arrow) & level IIb (arrow) left. (**B**) T2-weighted FFE shows clear negative contrast in corresponding SLNs, as a result of SPIO uptake. (**C**,**D**) Sagittal and coronal reconstruction of (**A**) shows the SLN in level IIa left (dotted arrow). (**A**–**D**) Informed consent has been obtained from this patient.

**Figure 5 cancers-12-03055-f005:**
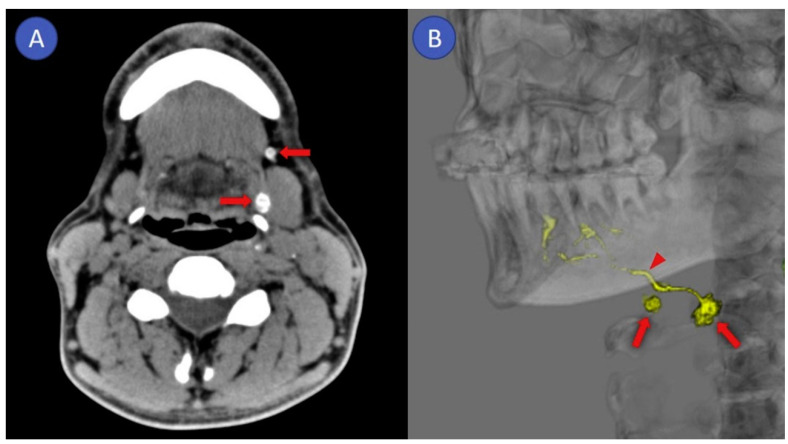
Computed tomographic lymphography: (**A**) axial image, (**B**) 3D image. Arrows: sentinel lymph node; arrowhead: lymphatics [27]. Figure used with permission of Elsevier©, permission license number: 4807630528815.

**Figure 6 cancers-12-03055-f006:**
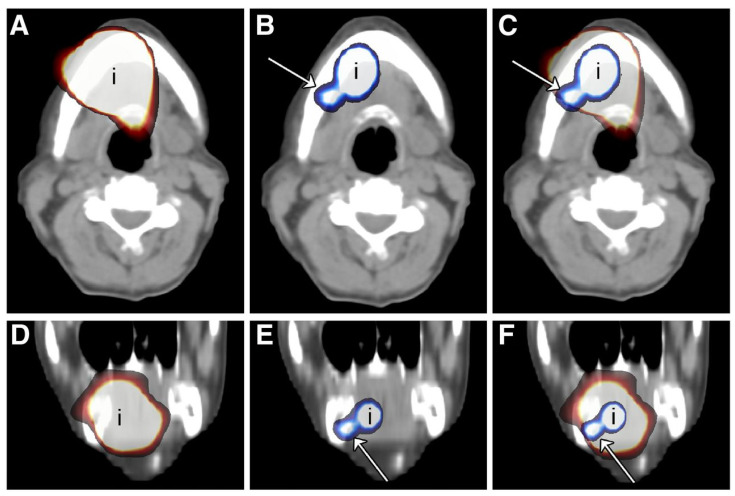
(**A**,**D**) Axial (**A**) and coronal (**D**) SPECT-CT image of injection site (i) of patient 1, i.e., floor of mouth, in which only a large hot spot from injection site could be visualized. (**B**,**E**) PET-CT image of injection site of same patient in which level IB lymph node (arrow) clearly could be identified. (**C**,**F**) Fused SPECT and PET-CT images showing that lymph node visualized on PET-CT is hidden behind large hot spot on SPECT-CT images [33]. This research was originally published in JNM [33]. Figure used with permission of original authors. ©SNMMI.

**Figure 7 cancers-12-03055-f007:**
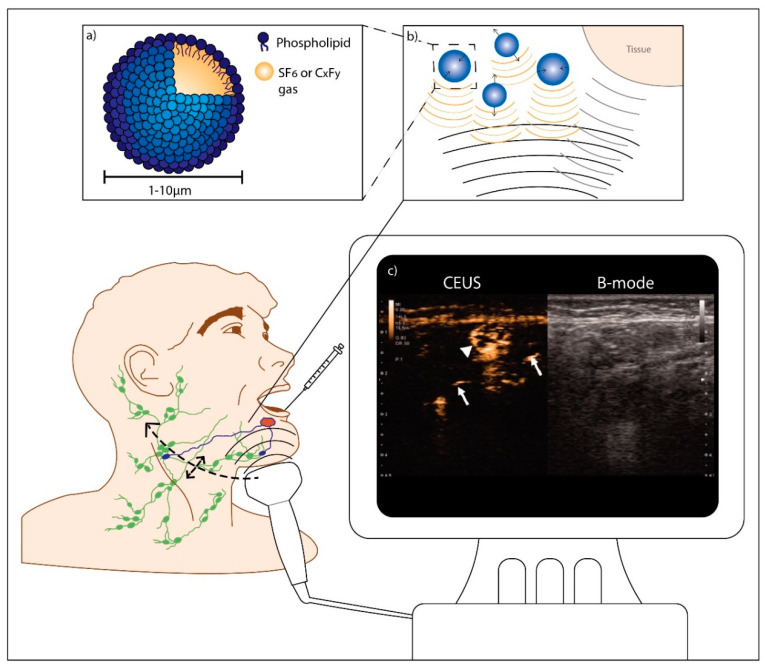
Contrast-enhanced lymphosonography in oral cancer. After microbubble injection at one or multiple peritumoral locations, contrast-enhancement of the injection site is visualized in ultrasound contrast mode. Using real-time imaging, the transportation of the microbubbles through lymphatic vessels may be followed until they accumulate in the sentinel lymph nodes. Subsequently, the neck is scanned for additional contrast-enhanced lymph nodes. Contrast-enhanced lymph nodes can be either marked for surgical resection or directly subjected to biopsy or aspiration cytology. Peritumoral injections can be repeated if necessary. (**a**) Schematic representation of a microbubble; (**b**) Principle of contrast-enhanced ultrasound (CEUS): oscillating microbubbles produce strong nonlinear scattering which can be distinguished from scattering by the surrounding tissue; (**c**) Contrast-enhanced ultrasonography with Sonazoid. On the left half is a contrast-enhanced image, and on the right is the B-mode image. Contrast-enhancement of sentinel lymph nodes (SLNs) (arrowheads) was observed concomitant with lymphatic ducts (arrows) draining the nodes. (Adapted with permission from [35], copyright 2019 Taylor & Francis Group: license number: 4810090088685).

**Table 1 cancers-12-03055-t001:** Reported diagnostic accuracy and detection rate of sentinel lymph nodes per technique.

Technique	Source	Tracer	Number of Studies	Sensitivity	NPV	SLN identification in (%) of Patients
Conventional lymphoscintigraphy & SPECT-CT	γ-ray	γ-emitting [^99m^Tc]-labelled radiotracer (e.g., [^99m^Tc]-nanocolloid)	*n* = 66	87% [16]	94% [16]	−
MR Lymphography (Gd^3+^)	Radio-wave	Paramagnetic (Gd^3+^) contrast agent (e.g., gadobutrol)	*n* = 1	91% [22]	93% [22]	100% [22]
MR Lymphography (SPIO)	Radio-wave	Superparamagnetic (iron oxide) contrast agent (e.g., Resovist, Magtrace)	*n* = 2	NR	NR	100% [23,24,25]
CT Lymphography	X-ray	Iodine contrast agent (e.g., iopamidol, lipiodol)	*n* = 6	56–80% [26,27,28,29]	82–96% [26,27,28,29]	89–96% [26,27,28,29,30,31]
PET lymphoscintigraphy	β^+^-decay (γ-rays)	Positron emitting isotope [^89^Zr, ^68^Ga, ^18^F]-labelled radiotracer (e.g., [^68^Ga]-tilmanocept)	*n* = 2	67% [32]	67% [32]	100% [32,33]
Contrast-enhanced lymphosonography	US-wave	Microbubbles (e.g., SonoVue, Sonazoid)	*n* = 2	NR	NR	80–92% [34,35]

NPV; negative predictive value, SLN; sentinel lymph node, SPECT-CT; single photon emission computed tomography-computed tomography, ^99m^Tc; technetium-99m, MR; magnetic resonance, Gd^3+^; gadolinium, NR; not reported, CT; computed tomography, PET; positron emission tomography, ^89^Zr; zirconium-89, ^68^Ga; gallium-68, ^18^F; fluorine-18, US; ultrasound.

**Table 2 cancers-12-03055-t002:** Merits and drawbacks per technique.

Technique	Advantages	Drawbacks
Conventional lymphoscintigraphy & SPECT-CT	Widely investigated and implemented Allows intraoperative localization of depicted SLNs Differentiation in intensity of radioactive signal Allows (intraoperative) differentiation between SLNs and HENs	Subject to shine-through phenomenon Requires nuclear facilities Low spatial resolution (~5 mm) Poor soft tissue contrast
MR Lymphography (Gd^3+^)	High spatial resolution (~1 mm) High signal-to-noise ratio and few artifacts Accurate anatomical detail Eliminates shine-through phenomenon Visualization lymphatic vessels May facilitate more targeted radiotherapy No nuclear facilities required Free of radiation exposure	Lacks intraoperative localization of depicted SLNs Rapid lymphatic transportation tracer No retention of tracer in SLNs Gd^3+-^based contrast agents not registered for lymphography
MR Lymphography (SPIO)	High spatial resolution (~1 mm) Accurate anatomical detail Allows intraoperative localization of depicted SLNs Eliminates shine-through phenomenon May facilitate more targeted radiotherapy No nuclear facilities required Free of radiation exposure	Limited clinical experience in OSCC Retention in SLNs depends on particle size Excess amounts of iron leads to signal voids Negative contrast may confound effectivity SLN detection Local inflammation following administration Metal elements interfere with magnetometer
CT Lymphography	High spatial resolution (~0.5 mm) High temporal resolution Eliminates shine-through phenomenon Visualization lymphatic vessels Visualization of lingual SLNs May facilitate more targeted radiotherapy No nuclear facilities required Widely available and low costs	Lacks intraoperative localization of depicted SLNs Rapid lymphatic transportation tracer No retention of tracer in SLNs Prone to artifacts Poor soft tissue contrast
PET lymphoscintigraphy	High spatial resolution (~2 mm) High temporal resolution Diminishes shine-through phenomenon Visualization lymphatic vessels Visualization of lingual SLNs Differentiation in intensity of radioactive signal Can be performed with known radiotracers Tri-model agent: IRD-800CW-[^68^Ga]-[^99m^Tc]-tracer Allows intraoperative localization of depicted SLNs	Requires nuclear facilities Poor intraoperative localization of SLNs with PET-probe Poor soft tissue contrast
Contrast-enhanced lymphosonography	Good safety profile of microbubbles High spatial resolution (~0.5 mm) High temporal resolution and real-time imaging Eliminates shine-through phenomenonPossibly no uptake of microbubbles in HENs Can be combined with USgFNA May be extended to other head and neck sites Widely available and low costs Free of radiation exposure	Limited clinical experience in OSCC Suspected low reproducibility High operator dependency Rapid lymphatic transportation tracer Challenging to mark SLNs for biopsy

SPECT-CT; single photon emission computed tomography—computed tomography, SLN; sentinel lymph node, HEN; higher echelon node, MR; magnetic resonance, Gd^3+^; gadolinium, SPIO; superparamagnetic iron oxide, OSCC; oral squamous cell carcinoma, CT; computed tomography, PET; positron emission tomography, IRD; infrared dye, ^68^Ga; gallium-68, ^99m^Tc; technetium-99m, USgFNA; ultrasound guided fine-needle aspiration.

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
