# Peer review of "New Developments in Imaging for Sentinel Lymph Node Biopsy in Early-Stage Oral Cavity Squamous Cell Carcinoma"

_cancers, 2020, doi:10.3390/cancers12103055_

Round 1

Reviewer 1 Report

Dear authors,

Please allow me a few comments:

  • The overall reading is sometimes long: are there ways to cut down the extent of the text? It may increase the readability of the manuscript.
  •  
  • Figures and tables are nicely chosen and very clear. I particularly like the Table 2 that summarizes the whole review.
  •  
  • Also I miss a bit a critical discussion about SLNB being a valid technique or not. What are the indications to perform SLNB? Mention the complex and rich lymphatic drainage system (looking at Figure 1 for example) compared to the skin or breast for example. What is the advantage of performing SLNB vs. neck dissection? Some countries like the USA are very restrictive about routinely doing SLNBs and a lot of people still need to gain confidence in using SLNB. Today still studies are being performed on this topic.
  •  
  •  With the title you focus on early stage OSCC, but why not perform SLNB in other areas of HNSCC? Don't we have good arguments to do SLNB in easily accessible cT1-T2 oropharyngeal cancers? What are the limitations?
  •  
  • In the conclusion could you give an indication/advise about what technique is the most promising? How will the future look? A combined approach?
  •  
  • The last sentence is of outmost importance in my opinion, moving away from one size fits them all to personalized medicine. I think this could be stressed more in the manuscript. Understand the cancer biology to improve cancer care.

Reviewer 2 Report

In this manuscript, the authors reviewed the diagnostic accuracy, the relative merits, disadvantages and potential applications of the novel SLNB imaging techniques for early-stage OSCC, including MR, lymphography, CT lymphography, PET lymphoscintigraphy and contrast-enhanced lymphosonography. As a review article, this manuscript was well designed and written. But it will be more helpful if the authors could discuss the differences between the animal/preclinical studies and the clinical studies.

Round 2

Reviewer 1 Report

Thank you for your thoroughly and in-depth point to point reply.